# Applying an Artificial Neuromolecular System to the Application of Robotic Arm Motion Control in Assisting the Rehabilitation of Stroke Patients—An Artificial World Approach

**DOI:** 10.3390/biomimetics8050385

**Published:** 2023-08-24

**Authors:** Jong-Chen Chen, Hao-Ming Cheng

**Affiliations:** Information Management Department, National Yunlin University of Science and Technology, Douliu 640, Taiwan; m11123010@yuntech.edu.tw

**Keywords:** rehabilitation, computational intelligence, evolutionary learning, self-organizing learning, robotic arm

## Abstract

Stroke patients cannot use their hands as freely as usual. However, recovery after a stroke is a long road for many patients. If artificial intelligence can assist human arm movement, it is believed that the possibility of stroke patients returning to normal hand movement can be significantly increased. In this study, the artificial neuromolecular system (ANM system) developed by our laboratory is used as the core motion control system to learn to control the mechanical arm, produce similar human rehabilitation actions, and assist patients in transiting between different activities. The strength of the ANM system lies in its ability to capture and process spatiotemporal information by exploiting the dynamic information processing inside neurons. Five experiments are conducted in this research: continuous learning, dimensionality reduction, moving problem domains, transfer learning, and fault tolerance. The results show that the ANM system can find out the arm movement trajectory when people perform different rehabilitation actions through the ability of continuous learning and reduce the activation of multiple muscle groups in stroke patients through the learning method of reducing dimensions. Finally, using the ANM system can reduce the learning time and performance required to switch between different actions through transfer learning.

## 1. Introduction

Stroke is a common disease in recent years, and there is a trend of gradual increase. Typically, a person experiences a partial loss of physical function and impaired cognitive function after a stroke. After a stroke, patients must undergo active rehabilitation, especially during the first three months. The role of rehabilitation therapy is to stimulate the brain to replace damaged brain cells with other brain cells through some exercise so that the lost function can be restored to a certain extent.

Generally speaking, the problem of a stroke patient is that he (or she) can make some movements autonomously but accompanied by obviously coordinated movements. For example, when a patient wants to raise their arms, his (or her) shoulder blades will retract and lift significantly, and the elbows will bend. This is because the action of human beings is not a single strand of muscle but the result of the coordinated movement of multiple muscles between the agonistic muscle and the antagonistic muscle. Generally speaking, to complete a successful action, two things must happen simultaneously. One is that the agonist muscle must contract to start a move, while the other is that the antagonist muscle must relax to inhibit it. The brain coordinates these movements to ensure muscle groups do not accidentally clash. To achieve this, it must send inhibitory or excitatory signals to the correct muscle groups so that they contract synchronously and efficiently. After a stroke, however, a patient’s brain’s ability to send the proper signals to muscle groups may be impossible, resulting in difficulty activating a single muscle group (instead, multiple muscle groups may be simultaneously activated).

Even worse, when the signal between the brain and the muscles malfunctions, it can cause the muscles to contract or spasm involuntarily in what is known as “abnormal coordinated movements”. Generally speaking, the best way to overcome “abnormal coordinated movements” after a stroke is through the repeated practice of correct rehabilitation movements. However, there is no question that this can be difficult to achieve when the patient’s condition does not improve gradually. In this case, if the patient is constantly asked to continue the rehabilitation exercise, the patient may be very distressed and eventually give up. The best approach is through the diagnosis of the physiotherapist and occupational therapist, who formulate a rehabilitation treatment plan that suits a patient’s individual needs. With a rehabilitation treatment plan, how to practice it is very important. There is no doubt that the help of a therapist (especially hands-on treatment) is crucial. However, hands-on treatment by a therapist is quite labor-intensive. Some scholars suggest using some instruments to drive the patient’s limbs to make continuous movements to assist in the treatment [1]. However, these early instruments did not provide the patient with any feedback information for reference and improvement. With the advancement of technology, the current design of robots not only excels in high precision, high reproducibility, and excellent reusability but can also provide multiple types of feedback information to users. It is generally accepted that the assistance of rehabilitation robots in implementing rehabilitation programs for patients may have the following three benefits. First, it can reduce the burden on the therapist and prevent sports injuries. Second, it can provide patients with a stable quality of treatment while performing repetitive, continuous, monotonous movements. Thirdly, a series of quantitative indicators can be systematically established to evaluate the degree of recovery of patients by obtaining the patient’s status and force data from different angles through intelligent robots.

In recent years, some have emphasized the so-called “adjuvant therapy strategies”. The idea is to use the robot to simulate the function of the therapist’s movements, such as giving the patient the right amount of resistance and help at the right time. Generally, this practice promotes the recovery of the patient’s health. When the patient cannot complete the required actions independently, the intelligent robot will use assisted force similar to that of a clinical physiotherapist [2,3,4,5,6,7,8]. An alternative to the use of assistive force is the use of electrical muscle stimulation [9,10]. However, which is the best way to assist the patient is still an open question [11].

The most commonly used current robot for assisting stroke patients is an end-point fixation system, such as MIT-Manus [12], which fixes the distal part of the patient to guide the desired motion. In 1998, the scholar Krebs designed a planar action robot arm (MIT-MANUS) to allow patients to move through the robot arm and use the information from the screen to assist their hands in acting successfully. Cozens [13] proposed a simple single-axis torque motor robot system in 1999 and proved that it is helpful for the upper limb rehabilitation training of patients with spasticity and muscle weakness. Hesse et al. [14,15] developed a pair of hands to control the robotic Bi-Manu-Track to improve forearm pronation/supination and wrist flexion coordination. In 2005, Colombo et al. [16] developed a shoulder–elbow robot with 2 degrees of freedom and a wrist robot with 1 degree of freedom (DOG) to assist patients with hand movements. The robots designed in the above studies mainly use the patient’s end contact to make different actions.

A group of scholars [17,18,19,20] pointed out that the upper limbs of patients after stroke often have so-called “abnormal coordinated movements”. These movements include shoulder external rotation and abduction, forearm elbow flexion, forearm supination or internal rotation, and flexion of the forearm and wrist. For example, when a patient tries to move the affected arm, his (or her) shoulders are raised and elbows bent. At present, some robot-assisted systems have been proven to be helpful to some extent in the upper-limb treatment of stroke patients. However, there may still be considerable variation in the ‘coordinated abnormality’ presented by different patients. Even the same patient may act differently in various situations. Aimed at these different situations, this study wishes to explore the above questions hypothetically, in the so-called “artificial world”. Our approach is to build up the action database systematically to a certain extent. Then, we use a brain simulation system with a self-organizing ability to establish the relationship between these actions and the possibility of action conversion.

## 2. The ANM System

As we all know, the main design of a digital computer is to provide a friendly programming environment. People first designed some symbols to express meaningful information. These symbols are manipulated by a set of rules that operate on them. The final result derived (or deduced) is interpreted by people later as the information people want to explore. Through these symbols and rules, people can use highly structured designs to find many solutions to problems. However, the meanings represented by these symbols are bridged by people’s abstract thinking, and the rules that operate on these symbols are the functions that they present. In this design, small changes (symbols and rules) can lead to significant changes in the results. Thus, a programmable system may frequently need to be redesigned due to minor changes in requirements. This problem refers to the inability to connect a symbol (word or vocabulary) with what it represents, the so-called symbol grounding problem [21]. It is difficult for a computer to understand what people mean by “meaning”, despite the “wisdom” it can generate. 

Biological systems do not have the so-called symbolic problem that the structure of biological systems is closely related to their functions. Autonomous learning is a fundamental property and the essence of biological systems. The origin of the ANM system, established by the team early, is to capture the possible information processing between neurons and inside neurons in the brain to capture the gradual structure/function capability of biological systems. These properties are crucial in the problem domain of robotics for rehabilitation assistance. On the one hand, each patient’s needs are different. The system must have long-term learning ability. That is, in a complex and complicated learning problem domain, the system should have the capability of long-term continuous learning. At the same time, when faced with changes in patient needs (that is, facing a moving or changing problem), the system must show fairly good adaptability in dealing with the changes.

### 2.1. ANM System Overview

The ANM system mainly uses the internal dynamics of information-processing neurons to simulate the capability of integrating different spatiotemporal signals. The motivation for this hypothesis stems from various physiological evidence that the cytoskeleton inside actual neurons performs a function of information integration. These neurons are then manipulated (controlled) by another group of neurons with a memory function, called control neurons, to complete the assigned task. All information-processing and control neurons form the central processing subsystem are shown in Figure 1.

### 2.2. The Processing Mechanism of an Information-Processing Neuron

This study captures information processing in actual neurons and hypothesizes that its internal dynamics occur in the cytoskeleton. The cytoskeleton can be imagined as a small network of input/output converters, just like the acceptance/response mediator (perception-action mediator) mentioned in physiology.

When external signals are transmitted to the cytoskeleton, a specific type of signal flow will occur (the components of the cytoskeleton will determine the type of signal). When different types of signals converge somewhere in the cytoskeleton, it may affect the enzymes in the cell to open or close the ion channels they control. The switching of these ion channels leads to an increase in the electrical potential inside the neuron. When the potential of the neuron rises to a certain level, the neuron will be triggered.

In this study, there are four types of cytoskeleton elements (Figure 2), namely, basic constituent molecules (microtubule, microfilament, neurofilament, denoted as C1, C2, and C3, respectively), readin enzymes, readout enzymes, and microtubule-associated proteins (MAPs), as described below:i.Readin enzymes: The function of readin enzymes in the cytoskeleton is to convert the signals transmitted to the neuron cell membrane into cytoskeletal signals. Its presence or absence determines whether signals sent to the neuron’s cell membrane are translated into cytoskeletal signals.ii.Readout enzymes: The function of readout enzymes is to convert cytoskeletal signals into external output signals transmitted to other neurons. However, only when a particular combination of signals converges where the readout enzyme is present is it sufficient to activate the readout enzyme (indirectly, it prompts the neuron to fire). Thus, the presence or absence of the readout enzyme will determine whether the neuron fires.iii.Constituent molecules: The function of these constituent molecules is to transmit information. When the readin enzyme converts the signal outside the cell membrane into the cytoskeletal signal, it will flow along the constituent molecules of the same type. We assume that there are three different types of constituent molecules. Each type of molecule has its transmission speed. Different types of molecule types influence each other to varying degrees. Changing these types of molecules will affect the type of signal. Figure 3 illustrates how the signal is transmitted. In this study, a two-dimensional circular lattice was used to simulate the signal flow on the cytoskeleton. When the signal reaches the edge of the annular lattice, it will pass to the other end of the annular space (as shown in Figure 3).iiii.MAP: The primary function of MAP is to connect the different basic components of the cytoskeleton and to be responsible for the transmission (influence) of signals between different types of molecules. When a cytoskeletal signal is transmitted to a site with a MAP, it activates the constituent molecules at the other end through the MAP. There are two possible scenarios for this activation. The first is to change the state of the constituent molecules at the other end, and the second is to promote the initiation of new cytoskeletal signals by the constituent molecules at the other end. Thus, changing the presence or absence of a MAP will determine whether the different constituent molecules can influence each other.

### 2.3. Control Neurons

The role of control neurons is to select (control) a group of information-processing neurons with internal dynamics, which will be trained (through evolutionary learning) to be specific input/output processors. Control neurons are divided into high-level control neurons and low-level control neurons. High-level control neurons are responsible for selecting (controlling) lower-level control neurons (this control combination will change as the learning time increases). The lower-level control neurons are responsible for manipulating (controlling) those groups of information-processing neurons with similar cytoskeletal structures (this control combination is fixed). Figure 4 shows the connection between control neurons and information-processing neurons.

### 2.4. How the ANM System Works?

The ANM system has a multi-level architecture. The evolutionary learning mechanism used in this study is similar to Darwin’s “mutation-selection”-type search method (Figure 5). The evolutionary learning mechanism is divided into two stages. The first stage is at the level of control neurons, and the second stage is at the level of information-processing neurons. The training of the system is conducted through alternately training the system in these two stages. We first train the system for a certain amount of time in the first stage of learning. Then, we train the system for a while in the second stage of learning. The whole learning process is to repeat the above-mentioned different stages of learning until the training ends. We note that the purpose of the first stage of learning is to generate a suitable combination of information-processing neurons, while the second stage is to modify the information-processing neurons selected. The learning steps at each stage can be summarized into the following three steps:i.Evaluation: evaluate the performance of each sub-network according to the experimental requirements, and find the best sub-network.ii.Copy: copy the variable settings of the best subnet to other subnets.iii.Variation: each subnet randomly mutates the original setting.

## 3. Application Domain

The data used in this study were generated using what is known as an “artificial world”. The approach is to systematically build up a database of actions and then change some of the data in a partially random fashion. In this study, we use the V-REP simulation system to design a humanoid robotic arm with 7 degrees of freedom (as shown in Figure 6), each controlled by a motor. We first use the V-REP simulation system to establish the trajectory data of a single movement and then generate the trajectory data of relatively complicated movements. Finally, we partially and randomly change the above trajectory data, assuming that each stroke patient cannot accurately act.

We note that the above data generated with an artificial world approach are not the movement trajectories of humans. Without the inherent limitation of human data, the artificial world approach provides a comparatively large data set for us to perform various functions. The purpose is to demonstrate that the ANM system (an artificial brain-like system) possesses good learning and adaptable capability. Under such considerations, the authenticity of data may be insignificant, but rather it is significant whether a successful transduction mechanism exists. We will call it a “success” if autonomous learning can make the appropriate transduction.

### 3.1. Trajectory Data for a Single Movement

For a person with healthy hands, they can move freely under the brain’s instructions. In the first experiment, we train the ANM system to learn the four actions (shrug, swing, side-lift, and up-lift) that healthy people can perform freely, as shown in Figure 7. Our simulation results show that when we turn the M1 motor 25 degrees, the M2 motor 40 degrees, the M3 motor 100 degrees, and the M4 motor 150 degrees, we can simulate human-like shrug, swing, lateral lifting, and upward movements, respectively. For each of the above actions, we fetch the end position (X, Y, Z coordinate values) of the robot arm every 50 microseconds during the execution of the action. Finally, we link each end position coordinate throughout the motion to create the goal trajectory. To facilitate the visualization of the two-degree space, we give the three-axis trajectory as three two-axis trajectory diagrams (as shown in Figure 8).

### 3.2. Trajectory Data of Complex Movements

In the data generated in the previous section, we established trajectory data for four basic actions (shrug, swing, side-lift, and up-lift). It is easy for people with healthy hands to perform any combination of these four movements. In terms of performing two out of the four actions simultaneously, there are six possible combinations. Likewise, there are four possible combinations if three of these four actions are performed simultaneously. Of course, the most complicated action is to perform these four actions simultaneously.

### 3.3. Data on Various Action Combinations for Different Patients after Stroke

The literature tells us that stroke patients often have “abnormal coordinated movements” of the upper limbs. This study hypothesizes that stroke sufferers have difficulty activating a single group of muscles and instead will produce a simultaneous activation of multiple groups of muscles. In this study, the above data were changed in a local random way to form the data of different stroke patients.

### 3.4. Input/Output Interface (Linkage to the ANM System)

All the information-processing neurons of the ANM system are equally divided into three groups according to the classification of the three axes. The firing behavior of each group of information-processing neurons represents the manipulation control of the robotic arm of a specific axis (Figure 9). In the current implementation, we use the time difference between two adjacent firing neurons of the same group to describe the degree of actuation of this axis; that is, there is an S-shaped waveform-like relationship (Equation (1)) between the time difference and the degree of actuation. The magnitude of this degree of motion will gradually decrease over time. This study also assumes that the relationship between the extent of the attenuation and the elapsed time is similar to an S-shaped waveform. It weakens slowly at first, but, over time, it weakens faster. Before the drive completely wears off, the new waveform from another firing neuron is added to the old waveform to form a larger waveform. The entire driving waveform represents the driving of an axis at the finger-end of the manipulator.

If a new waveform is generated shortly after one waveform, the two waveforms will be processed in a tandem superimposed manner. A double-peak waveform is formed at the two peaks, and the overlapping area in the middle will be superimposed. The waveforms generated by the firing behavior of the neurons in the same group were superimposed to form a relatively large waveform. The ANM system superimposes (transduces) all the waveforms generated from the input records into time-series data of hand activities. The final waveform (the time-series data) will be contrasted with the time-series data of healthy subjects. The fitness is measured as the dynamic time-warping distance (to be referred to as DTW) between these two time-series data, as shown in Equation (2). The smaller the DTW distance, the better the system’s fitness.
(1)$$\mathrm{Degree}\;\mathrm{of}\;\mathrm{actuation}=(\frac{1}{1+e\left(-2\times t\right)}\;-\;0.5)\;\times \;2\;\times \;90$$
(2)$$\mathrm{DTW}=\sum_{i}\left|\sum_{j=1}^{50}\left(V_{ij}-A_{ij}\right)\right|$$
where $V_{ij}$ and $A_{ij}$ represent the target trajectory generated by the V-REP system and the trajectory generated by the ANM system, respectively, *i* = *x*-axis, *y*-axis, and *z*-axis. 

## 4. Experiments

We know that in nature, continuous evolutionary learning is an essential condition for the survival of biological systems, especially in an unknown environment. Five experiments are conducted in this research. The first part focuses on the learning ability of the system. The second part lets the system learn actions with reduced degrees of freedom. This experiment considers how to reduce the number of muscle groups stimulated. The third part explores the adaptive learning ability of the system, meaning the ability to adapt to a moving (or changing) problem. The fourth part investigates how machine learning knowledge learned from a task is re-used to boost the learning performance on another related task. The fifth part examines how a learned system copes with environmental change, assuming patients have different types of strokes.

### 4.1. Perpetual Learning

In this experiment, we first let the ANM system learn the trajectory curves of the above four actions. The learning results are shown in Figure 10. In the early stage of learning, the gap between DTW distances increases and is relatively large, but, in the later stages, this range becomes smaller and smaller. Most importantly, the system still showed continuous improvement, even in the late stage of learning. The main reason for the ANM system is the gradual transformation of structure/function. This tests the system’s learning ability for an intelligent system. In particular, when our system changes the difficulty of the problem to a considerable extent, the system must show the capability of continuous improvement. This experiment also let the ANM system learn the trajectory curves in two, three, and four DOGs. We note that the learning of the ANM system is terminated at cycle 300, and the results are shown in Table 1. The improvement in the regression coefficient R2 value in Table 1 shows that the ANM system presents a good learning effect. This study also allows the system to continue learning to about 2000 generations. The results show that the system can show continuous improvement for each action. However, to explore the different information processing of the ANM system more extensively, the experimental results in this part of this paper stop at the place where the number of learning times is 300 generations. This is because most of the neurons in the ANM system have internal dynamic processing functions; the computational time required to simulate this type of data processing with a digital computer is relatively large (especially using a sequential processing machine to simulate a parallel processing system).

### 4.2. Adaptive Learning

The motivation for this experiment was to think from the perspective of the rehabilitation of stroke patients. As mentioned earlier, a stroke patient often has a “coordinated movement disorder” in the upper limbs, that is, the simultaneous activation of multiple muscle groups. If the patient can reduce the possibility of stimulating several groups of muscles simultaneously, the patient is relatively more able to perform the desired action.

As far as a person with healthy hands is concerned, they can perform the actions they want to perform according to their free will. However, a patient after a stroke might have abnormal coordinated movements. The following experiment is motivated by how to help the rehabilitation of stroke patients by reducing the possibility of several groups of muscles being activated simultaneously when a patient wants to perform some simple movements. The approach taken in this study is to have the ANM system learn to reduce actions with relatively more degrees of freedom to actions with relatively fewer degrees of freedom. We train the system to move simultaneously from four to three degrees of freedom, then to two degrees of freedom, and finally to one degree of freedom. In addition to gradually reducing the number of activities one by one, another approach facilitates more activity change at a time. 

Table 2 shows that the system can successfully assist patients to move simultaneously from four to three degrees of freedom. From the perspective of assisting stroke patients, the system can help to reduce the possibility of stimulating several muscle groups simultaneously (that is, the patient is relatively more able to perform the desired movement). Table 3 shows the improvement rates from moving simultaneously in four to three degrees of freedom, then to two degrees of freedom, and finally to one. Table 4 shows the improvement rates from moving simultaneously in three degrees of freedom to two degrees of freedom and finally to one degree of freedom. Table 5 shows the results of the improvement rates from moving simultaneously in two degrees of freedom to one degree of freedom. This implies that the ANM system can help patients improve their simultaneous movement problems.

### 4.3. Learning in Moving Problem Domain

The experiment is performed to test the adaptability in confronting domain changes. The approach adopted is that we first let the system learn a set of actions for a while, and then the learned system is asked to learn another set of steps. The whole experiment has two purposes. The first purpose is to test the system’s performance against changing systems (that is, to understand the system’s ability to adapt to changes in the problem). The second purpose is to investigate whether the system can reproduce the ability to adapt to changes in the problem domain through the previous learning results. From the perspective of rehabilitation, the goal of this experiment can be interpreted as first letting the patient learn a set of movements, and later the rehabilitation movements required are changed in order to know how the patient can adapt to the change through the help of the system. The experimental results are shown in Table 6, Table 7 and Table 8.

### 4.4. Transfer Learning

Transfer learning (TL) is a research field in which the machine learning knowledge learned from a task is re-used to boost the learning performance on another related task. This is one of the generalization capabilities that a learned system applies to a family of relevant problem domains. From the point of view of the intelligent system assisting patients in rehabilitation, it means whether it is possible to use the learning results of some previous rehabilitation actions to make it easier to learn different, but relevant, rehabilitation actions. Regarding transfer learning, two issues need to be addressed. The first issue is whether a system that leverages the performance of new job assignments is better than one that does not utilize transfer learning at the beginning of learning. The second issue is whether the system has a better effect on the performance of new job assignments than if it does not use transfer learning after using the same long learning time (for example, after learning 500 learning cycles) (as shown in Table 9).

### 4.5. Noise Tolerance

As mentioned earlier, stroke patients stimulate multiple muscle groups at the same time. Additionally, different stroke patients may have different levels of muscle group stimulation. In this study, the motion trajectories of the simultaneous stimulation of multiple muscle groups were varied in a local stochastic manner to shape stroke conditions in different patients. For each rehabilitation action, five different trajectories were generated. The method of this experiment is to take the results of the first part of the experiment (i.e., the system after long-term learning), then test it with the trajectories generated above and take the average of these five runs. The results are shown in Table 10.

## 5. Discussion

We know that biological systems have an excellent adaptability that allows them to continue operating (or operating) in an unknown or uncertain environment. An ambiguous or poorly defined problem domain belongs to a novel or partially unfamiliar territory, while a disturbed or changing problem domain belongs to a so-called uncertain or uncertain environment in statistical terms. Adaptability includes two abilities: learning and adaptation. It should be noted that adaptability is a broader term (or concept) than adaptation because it, more broadly, has self-regulation to correct errors. 

Adaptability is undoubtedly a highly desirable goal from the perspective of using robots in assisting patients’ rehabilitation. To establish a rehabilitation assistance system suitable for different people and different needs, an intelligent system must possess a rich learning capability to perform long-term continuous learning on a complicated learning problem and a considerable degree of plasticity for adaptation to different needs. Since each patient is injured differently, the need for an assistive robot differs. Customized design is an inevitable trend. An intelligent assistance system must be able to find the best answer to assist the patient in a self-correcting manner. In addition, the system must also have some degree of noise tolerance to cope with transient variation in the patient’s actuation because the method and strength of each patient’s movement are not exactly the same. Noise tolerance refers to the ability of a system to continue operating in a disturbed environment.

Adaptive learning is another important feature that an intelligent rehabilitation assistance system must have when there are permanent changes in the problem domain. From a rehabilitation point of view, this requirement can be interpreted as the patient having to learn new movements, i.e., changing the required rehabilitation movements. This aspect includes incremental or mobile changes in the requirements of rehabilitation movements. In this case, using the previously learned action experience to accelerate the adaptation to remember new actions (i.e., transfer learning) is essential.

## 6. Conclusions

Robotic rehabilitation is a method that achieves sensorimotor training with external mechanical assistance. The current development of assistive robots mainly integrates the most advanced aspects of artificial intelligence, medicine, biomechanics, and human factors engineering to meet the following two requirements. The first requirement is to produce precise control over the machine, and the second is to meet the needs of different customizations. However, these two requirements can sometimes take work to be satisfied simultaneously. The intelligent system acts as a bridge between the two requirements.

The difference between the traditional neural network and the ANM system proposed in this study lies in the information processing inside the neuron. The former is the learning result generated using mathematical calculations, while the latter emphasizes information processing inside neurons. The former must be completed through network information processing to simulate spatiotemporal information processing. In contrast, the ANM system used in this study can be expressed in terms of the internal dynamics of a single neuron. In other words, a single neuron in the ANM system can process spatiotemporal information sufficient to represent the traditional neural network. Because of this, we can capture the characteristics of gradual changes in biological structure/function by simulating the internal activities of neurons through information processing inside neurons. The experimental results in this study prove that it has perpetual learning capability and sufficient adaptability.

## Figures and Tables

**Figure 1 biomimetics-08-00385-f001:**
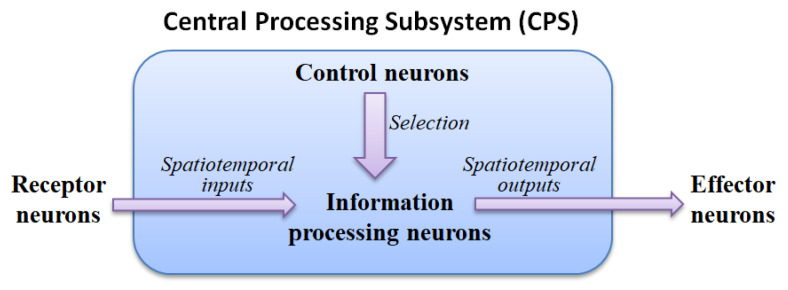
Central processing subsystem.

**Figure 2 biomimetics-08-00385-f002:**
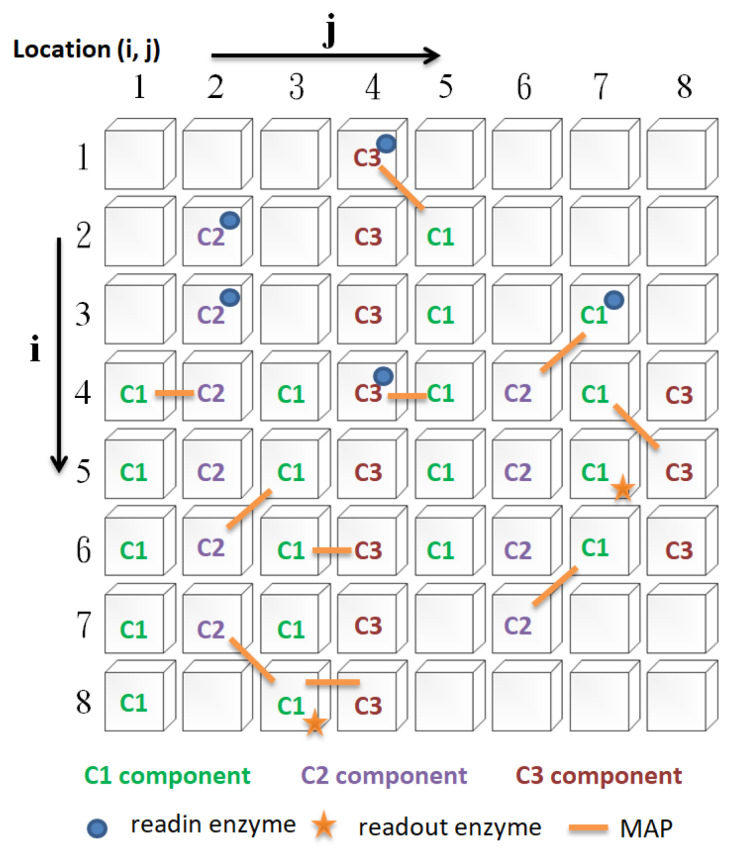
Cytoskeleton elements.

**Figure 3 biomimetics-08-00385-f003:**
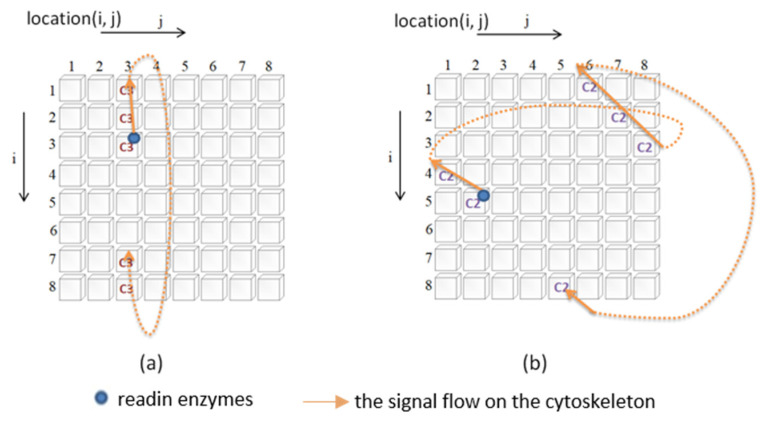
(**a**) An example of an upward flow signal on a circular grid. (**b**) An example of an upward-left signal flow on a circular grid.

**Figure 4 biomimetics-08-00385-f004:**
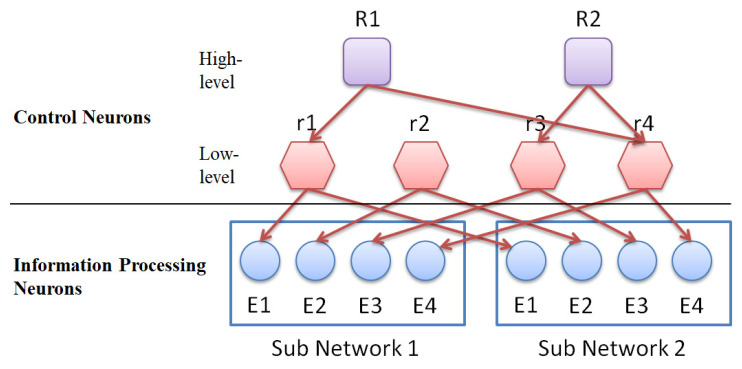
The connection between control neurons and information processing neurons.

**Figure 5 biomimetics-08-00385-f005:**
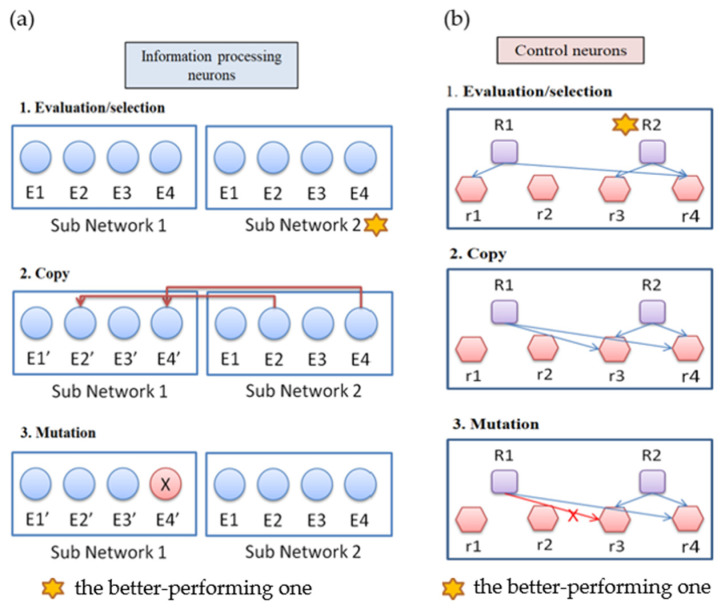
(**a**) Evolutionary learning at the level of information processing neurons. (**b**) Evolutionary learning at the level of control neurons.

**Figure 6 biomimetics-08-00385-f006:**
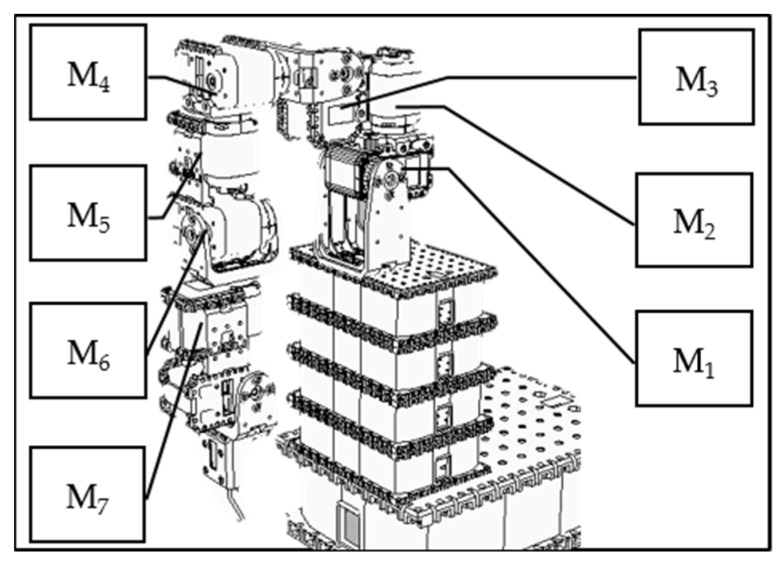
A humanoid robotic arm using a V-REP simulation environment.

**Figure 7 biomimetics-08-00385-f007:**
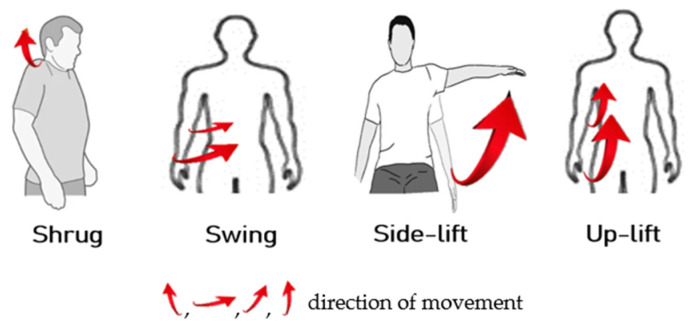
Four actions that healthy people can perform freely.

**Figure 8 biomimetics-08-00385-f008:**
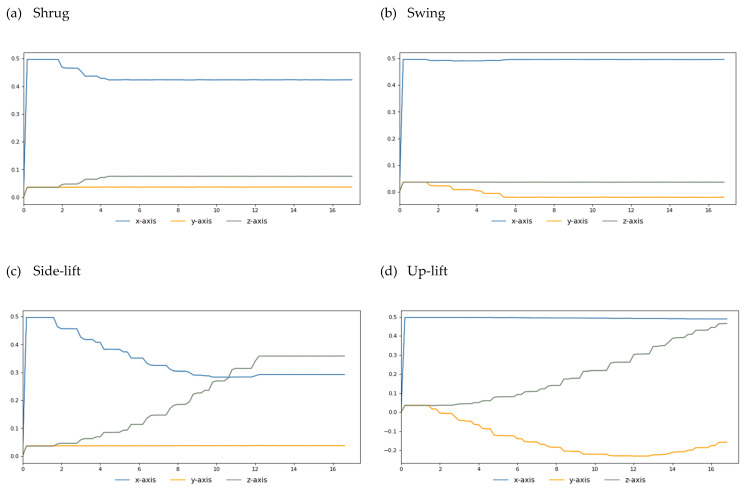
(**a**) Three biaxial trajectory lines represent the three-axial trajectory of the shrug action. (**b**) Three two-axis trajectory lines represent the three-axis trajectory of the swing action. (**c**) Three two-axis trajectory lines represent the three-axis trajectory of the side-lift action. (**d**) Three two-axis trajectory lines represent the three-axis trajectory of the up-lift action.

**Figure 9 biomimetics-08-00385-f009:**
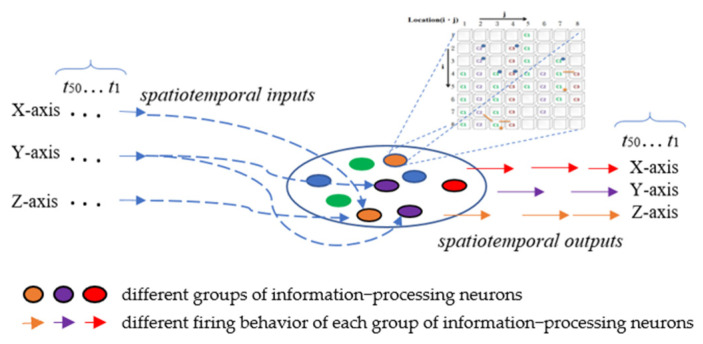
Input/output interface of the ANM system.

**Figure 10 biomimetics-08-00385-f010:**
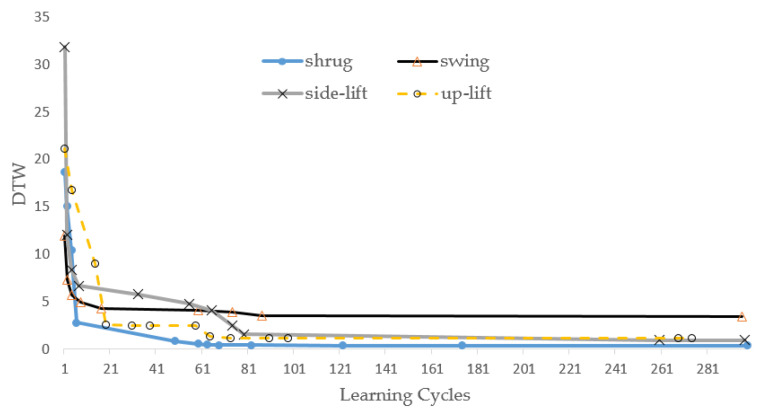
Learning results of each of these four actions.

**Table 1 biomimetics-08-00385-t001:** Each rehabilitation action’s DTW and R2 score at cycles 1 and 300.

Rehabilitation Action	DTW At Cycle 1	DTW At Cycle 300	R^2^ Score At Cycle 1	R^2^ Score At Cycle 300
Shrug, Swing, Side-lift, Up-lift	40.25	4.64	−12.63	0.69
Shrug, Swing, Side-lift	27.12	5.96	−18.31	0.55
Shrug, Swing, Up-lift	32.94	3.75	−18.95	0.63
Shrug, Side-lift, Up-lift	33.73	4.03	−17.85	0.70
Swing, Side-lift, Up-lift	35.89	4.19	−17.51	0.85
Shrug, Swing	28.92	4.39	−37.01	0.73
Shrug, Side-lift	28.29	6.44	−24.38	0.61
Shrug, Up-lift	29.96	5.01	−25.77	0.82
Swing, Side-lift	25.30	3.99	−16.19	0.44
Swing, Up-lift	37.29	5.62	−21.94	0.53
Side-lift, Up-lift	28.90	4.16	−33.43	0.60

**Table 2 biomimetics-08-00385-t002:** The improvement from four to three degrees of freedom.

	To	To Three Movements											
		Shrug, Swing, Side-Lift			Shrug, Swing, Up-Lift			Shrug, Side-Lift, Up-Lift			Swing, Side-Lift, Up-Lift		
From		Before	After	Rate	Before	After	Rate	Before	After	Rate	Before	After	Rate
Shrug, Swing, Side-lift, Up-lift		41.79	3.85	90.8%	45.51	3.82	91.6%	45.59	4.02	91.2%	49.26	4.75	90.4%

**Table 3 biomimetics-08-00385-t003:** The improvement from four to three degrees of freedom, then to two, and finally to one.

	To	To Three Movements				To Two Movements						To Single Movement			
From															
From Four Movements		Shrug, Swing, Side-Lift	Shrug, Swing, Up-Lift	Shrug, Side-Lift, Up-Lift	Swing, Side-Lift, Up-Lift	Shrug, Swing	Shrug, Side-Lift	Shrug, Up-Lift	Swing, Side-Lift	Swing, Up-Lift	Side-Lift, Up-Lift	Shrug	Swing	Side-Lift	Up-Lift
Shrug, Swing, Side-lift, Up-lift		90.8%	91.6%	91.2%	90.4%	93.8%	88.8%	92.7%	88.6%	90.4%	93.8%	92.8%	90.4%	88.5%	90.2%

**Table 4 biomimetics-08-00385-t004:** The improvement from three to two degrees of freedom and finally to one.

	To	To Two Movements						To Single Movement			
From											
From Three Movement		Shrug, Swing	Shrug, Side-Lift	Shrug, Up-Lift	Swing, Side-Lift	Swing, Up-Lift	Side-Lift, Up-Lift	Shrug	Swing	Side-Lift	Up-Lift
Shrug, Swing, Side-lift		84.2%	85.7%	91.4%	89.5%	87.3%	89.8%	90.7%	92.1%	90.7%	89.6%
Shrug, Swing, Up-lift		92.0%	91.4%	90.3%	91.7%	90.0%	88.6%	86.3%	91.3%	90.9%	93.6%
Shrug, Side-lift, Up-lift		92.0%	87.3%	91.4%	92.4%	89.3%	91.9%	91.3%	92.3%	86.0%	92.0%
Swing, Side-lift, Up-lift		91.2%	88.8%	93.2%	89.2%	90.1%	94.8%	90.0%	93.8%	85.0%	93.0%

**Table 5 biomimetics-08-00385-t005:** The rate of improvement from two degrees of freedom to a single degree of freedom.

	To	To Single Movement			
From					
From Two Movement		Shrug	Swing	Side-Lift	Up-Lift
Shrug, Swing		89.9%	87.1%	90.1%	87.0%
Shrug, Side-lift		85.9%	92.9%	84.7%	90.8%
Shrug, Up-lift		90.6%	95.4%	89.4%	88.9%
Swing, Side-lift		89.9%	90.2%	86.1%	93.2%
Swing, Up-lift		88.6%	92.2%	92.0%	91.7%
Side-lift, Up-lift		83.7%	86.5%	92.0%	87.1%

**Table 6 biomimetics-08-00385-t006:** The result of the adaptability in confronting one degree of freedom of domain changes.

	To	Shrug	Swing	Side-Lift	Up-Lift
From					
Shrug		−	90.1%	92.4%	86.6%
Swing		95.8%	−	86.9%	91.6%
Side-lift		94.2%	89.9%	−	89.9%
Up-lift		82.9%	89.8%	90.6%	−

**Table 7 biomimetics-08-00385-t007:** The result of the adaptability in confronting two degrees of freedom of domain changes.

	To	Shrug, Swing	Shrug, Side-Lift	Shrug, Up-Lift	Swing, Side-Lift	Swing, Up-Lift	Side-Lift, Up-Lift
From							
Shrug, Swing		−	87.3%	83.3%	83.4%	91.2%	89.2%
Shrug, Side-lift		88.9%	−	94.3%	93.7%	89.5%	91.2%
Shrug, Up-lift		89.3%	91.1%	−	92.6%	93.7%	89.1%
Swing, Side-lift		90.9%	87.7%	88.0%	−	90.1%	90.2%
Swing, Up-lift		89.6%	90.3%	92.4%	93.1%	−	87.9%
Side-lift, Up-lift		89.2%	90.7%	90.6%	89.9%	89.0%	−

**Table 8 biomimetics-08-00385-t008:** The result of the adaptability in confronting three degrees of freedom of domain changes.

	To	Shrug, Swing, Side-Lift	Shrug, Swing, Up-Lift	Shrug, Side-Lift, Up-Lift	Swing, Side-Lift, Up-Lift
From					
Shrug, Swing, Side-lift		−	91.6%	91.4%	88.4%
Shrug, Swing, Up-lift		95.4%	−	90.4%	93.1%
Shrug, Side-lift, Up-lift		91.9%	87.2%	−	91.5%
Swing, Side-lift, Up-lift		93.2%	93.2%	91.4%	−

**Table 9 biomimetics-08-00385-t009:** Compare the improvement rate of using transfer learning with not using transfer learning.

	At Cycle 1			At Cycle 500		
Rehabilitation Action	Without TL (DTW)	With TL (DTW)	Improvement Rate	Without TL (DTW)	With TL (DTW)	Improvement Rate
Shrug, Swing, Side-lift	30.11	12.75	57.7%	6.85	4.79	30.1%
Shrug, Swing, Up-lift	36.59	10.06	72.5%	4.92	5.24	−6.5%
Shrug, Side-lift, Up-lift	34.57	9.54	72.4%	10.97	5.80	47.1%
Swing, Side-lift, Up-lift	40.02	4.88	87.8%	5.94	4.31	27.4%
Shrug, Swing	28.73	17.72	38.3%	6.31	6.26	0.8%
Shrug, Side-lift	29.05	20.48	29.5%	7.35	7.48	−1.8%
Shrug, Up-lift	33.86	8.05	76.2%	6.18	5.28	14.6%
Swing, Side-lift	28.61	13.20	53.9%	6.55	6.10	6.9%
Swing, Up-lift	38.55	11.03	71.4%	5.57	5.24	5.9%
Side-lift, Up-lift	32.77	9.33	71.5%	5.98	4.56	23.7%
Shrug	29.17	19.69	32.5%	7.03	6.32	10.1%
Swing	31.27	17.52	44.0%	4.88	5.78	−18.4%
Side-lift	26.71	20.05	24.9%	7.38	7.18	2.7%
Up-lift	34.26	9.34	72.7%	4.09	4.33	−5.9%

**Table 10 biomimetics-08-00385-t010:** The improvement rate of noise tolerance at cycle one and cycle 500.

	At Cycle 1			At Cycle 500		
Rehabilitation Action	Without TL (DTW)	Random (DTW)	Improvement Rate	Without TL (DTW)	Random (DTW)	Improvement Rate
Shrug, Swing, Side-lift	30.11	20.18	57.7%	6.85	6.05	30.1%
Shrug, Swing, Up-lift	36.59	15.23	72.5%	4.92	5.06	−6.5%
Shrug, Side-lift, Up-lift	34.57	22.41	72.4%	10.97	6.10	47.1%
Swing, Side-lift, Up-lift	40.02	15.42	87.8%	5.94	4.83	27.4%
Shrug, Swing	28.73	23.20	38.3%	6.31	6.53	0.8%
Shrug, Side-lift	29.05	18.28	29.5%	7.35	5.81	−1.8%
Shrug, Up-lift	33.86	12.88	76.2%	6.18	5.46	14.6%
Swing, Side-lift	28.61	27.32	53.9%	6.55	5.92	6.9%
Swing, Up-lift	38.55	10.89	71.4%	5.57	5.45	5.9%
Side-lift, Up-lift	32.77	20.34	71.5%	5.98	6.59	23.7%
Shrug	29.17	11.77	32.5%	7.03	4.81	10.1%
Swing	31.27	11.31	44.0%	4.88	3.49	−18.4%
Side-lift	26.71	22.38	24.9%	7.38	5.43	2.7%
Up-lift	34.26	10.56	72.7%	4.09	4.53	−5.9%

## Data Availability

The data can be accessed through the link: https://reurl.cc/ZW05R3.

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
