# Peer review of "Applying an Artificial Neuromolecular System to the Application of Robotic Arm Motion Control in Assisting the Rehabilitation of Stroke Patients—An Artificial World Approach"

_biomimetics, 2023, doi:10.3390/biomimetics8050385_

Round 1

Reviewer 1 Report

Authors have carried out a relevant work and the paper could be considered for publication after the following minor corrections.

·       Could the authors include a description of the sub-figure. E.g. Figure 3. The title of Figure 3 should include a description of 3(a) and 3(b)

·       Figure 8 contains four images which should be labelled with a, b, c and d and the description included in the figure title for each. The same thing should be done any other place in the paper where the sub-figures were not labelled and described.

·       Instead of plotting the learning results for each action separately as shown in Figure 10. It would be better to have all the four as one plots for easy comparison.

Author Response

[Comment 1] Could the authors include a description of the sub-figure? E.g., Figure 3. The title of Figure 3 should include a description of 3(a) and 3(b)

[Response] It is corrected as suggested.

[Comment 2] Figure 8 contains four images that should be labeled with a, b, c, and d, and the description should be included in the figure title for each. The same thing should be done anywhere in the paper where the sub-figures were not labeled and described.

[Response] Figure 8 is corrected as suggested. Also, Figure 5 is corrected.

[Comment 3] Instead of plotting the learning results for each action separately, as shown in Figure 10. Having all four as one plot for easy comparison would be better.

[Response] All four learning curves are plotted together (It is corrected as suggested).

Reviewer 2 Report

The authors present a promising AI approach for stroke rehabilitation, the ANM system. The ANM system can learn arm movement trajectories, reduce muscle activation, and switch between actions quickly. This is a quiet interesting system. However, the authors only used the improvement rate to evaluate the ANM system (using transfer learning), which is not a comprehensive metric. A more comprehensive metric would be the R2 score metric, which takes into account both the improvement rate and the initial performance of the system based on prior "good movements" given by medical doctors. The authors should consider using the R2 score metric in future studies. Nevertheless, congrats for the study its well done, good designed and sounds good.

Author Response

The authors present a promising AI approach for stroke rehabilitation, the ANM system. The ANM system can learn arm movement trajectories, reduce muscle activation, and switch between actions quickly. This is quite an interesting system.

[Comment 1] However, the authors only used the improvement rate to evaluate the ANM system (using transfer learning), which is not a comprehensive metric. A more comprehensive metric would be the R2 score metric, which considers both the improvement rate and the initial performance of the system based on prior "good movements" given by medical doctors. The authors should consider using the R2 score metric in future studies. Nevertheless, congrats on the study it’s well done, well designed, and sounds good.

[Response] According to the valuable opinions of the review committee, the authors added the R2 value in Table 1.
